# Unlocking Lethal Dingo Management in Australia

**Louise Boronyak** [1,2,*], **Brent Jacobs** [2] and **Bradley Smith** [3]

1. Humane Society International Australia, Sydney, NSW 2107, Australia
2. Institute for Sustainable Futures, University of Technology Sydney, Ultimo, NSW 2007, Australia
3. School of Health, Medical and Applied Sciences, Central Queensland University, Adelaide Campus, Wayville, SA 5034, Australia
* Correspondence: lboronyak@hsi.org.au

**Abstract:** Adoption by livestock producers of preventive non-lethal innovations forms a critical pathway towards human and large carnivore coexistence. However, it is impeded by factors such as socio-cultural contexts, governing institutions, and 'perverse' economic incentives that result in a 'lock-in' of lethal control of carnivores in grazing systems. In Australian rangelands, the dingo is the dominant predator in conflict with 'graziers' and is subjected to lethal control measures despite evidence indicating that its presence in agricultural landscapes can provide multiple benefits. Here we explore the barriers to the uptake of preventive innovations in livestock grazing through 21 in-depth interviews conducted with Australian graziers, researchers, and conservation and government representatives. Drawing on Donella Meadow's leverage points for system change framework, we focus, primarily, on barriers in the 'political sphere' because they appear to form the greatest impediment to the adoption of non-lethal tools and practices. These barriers are then discussed in relation to characteristics of lock-in traps (self-reinforcement, persistence, path dependencies, and undesirability) to assess how they constrain the promotion of human–dingo coexistence.

**Keywords:** social-ecological systems; conflict; human-wildlife coexistence; dingo; livestock production; lock-in traps

## 1. Introduction

Coexistence with wildlife is increasingly being advocated as an important pathway to reduce threats to biodiversity by advocating tools and practices to mitigate the intentional killing of wildlife to improve animal welfare and conservation outcomes [1,2]. Coexistence is a behavioural state in which individual species live together in the same landscape at the same time and interact in mutually beneficial or reciprocal ways [3]. Human–carnivore coexistence in agricultural landscapes largely focuses on non-lethal tools and practices that aim to alter either human or wildlife behaviour to mitigate human–wildlife conflict and foster coexistence [4,5]. These include tools and practices that aim to prevent livestock predation but are non-lethal to wildlife, such as livestock husbandry (e.g., guarding and herding), enclosures (e.g., night pens, fencing), and predator deterrents (e.g., flashing lights and sounds) [1,6–8]. Some of these measures take advantage of carnivore neophobia and curtail learned behaviour, such as seeking out anthropogenic food sources, that can lead to conflicts [9]. Each of the tools and practices works in different ways and can be adapted to the local context (livestock type, terrain, local wildlife). Furthermore, preventive innovations can be used individually (e.g., guarding dogs) or in combination (e.g., dogs by day, pens by night). This reflects contemporary thinking about fostering human tolerance by reducing the costs of living alongside large carnivores and increasing the benefits of wildlife to shift interactions from conflict to coexistence in shared landscapes [10].

Preventive innovations serve as promising alternatives to lethal carnivore control. They aim to safeguard domestic livestock and large carnivores from harm, thus contributing to more ecologically sustainable animal-based agriculture [5]. The adoption of preventive innovations is occurring in niches across several countries [1,2,6,11–13]. In Australia,

non-lethal preventive innovations include livestock guardian animals [7–9,14], predator deterrents, low-stress livestock handling, buffer zones, and the removal of attractants such as carcasses [7,8]. These tools and practices can shift the focus from controlling dingoes to controlling dingo impacts, including methods aimed at protecting livestock [15,16]. However, the implementation of coexistence tools and practices is hampered by various cultural, practical, and political barriers, and their adoption is far from being 'mainstreamed'.

The dingo is a free-ranging member of the genus *Canis* and the largest mammalian predator in Australia. They are a highly adaptable generalist species that is present across the mainland. They have been present in Australia for at least 5000 years and are genetically, morphologically, behaviourally, and ecologically distinct from domestic dogs [17]. The dingo is perceived differently across Australia as an apex predator, icon, agricultural pest, and spiritual totem for Indigenous Australians. These narratives influence human interaction with dingoes. Dingoes are of cultural, social, and spiritual significance to a great many Australians, most significantly Indigenous Australians, who hold kinship ties and traditional knowledge about dingoes, other native animals, and the Australian landscape [18]. However, these positive values appear insufficient to overcome the dominant paradigm that narrows the perception of dingoes to agricultural pests.

Dingoes have been subject to targeted eradication attempts via various forms of lethal control (poison, shooting, trapping) since European settlement over 200 years ago [5]. This is a direct response to their actual and perceived threat to livestock production, namely sheep [19,20]. Management programs use the term 'wild dogs' as an umbrella term that encompasses 'pure' dingoes (with no modern dog ancestry), hybrid dingo-dogs, and feral dogs. This reflects the difficulty of distinguishing dingoes from their hybrids and the fact that most forms of control are indiscriminating and intended to remove all canids from the landscape regardless of ancestry. Recent genetic studies, however, have revealed wild canids in Australia are mostly dingo [21]. Often, the perspective and background of the person observing them influences how dingoes are described [22]. This explains why participants in this study may refer to dingoes as wild dogs. However, for the purpose of this paper, we consider these the same animal.

A growing number of cattle producers are starting to experience greater benefits by ceasing lethal control and learning to coexist with dingoes instead. They view the dingo is a valuable land management tool [23]. Evidence from graziers and ecologists indicates that stable family groups of dingoes can regulate the numbers of wild grazing animals [24,25]. This allows management and resting of total grazing pressure in paddocks, likely resulting in more abundant and higher-quality feed for livestock. This can lead to improved long-term landscape health and healthy and more resilient livestock, which result in better financial returns [24,26].

Coexistence between humans and dingoes has remained elusive because it requires a fundamental transformation in our relationships to and interactions with nature. Transformation is a deliberate attempt to promote a major, fundamental change in social-ecological systems [27]. Meadows' leverage point framework [28] and its iterations (e.g., O'Brien [29]) can be used to identify intervention points that foster transformation towards human–carnivore coexistence [30]. Intervention points are referred to as leverage points can be shallow or deep according to the type of influence they have on a system [28,31]. O'Brien refined Meadows' framework by grouping the 12 leverage points across 3 spheres: practical, political, and personal [29]. The notion of 'spheres' is used to reflect areas or domains that are an intrinsic part of a larger whole [29]. It enables a holistic view of the discrete elements and interrelationships of a system so that they can be addressed separately, thus providing a promising approach for conceptualising transformation in social-ecological systems [32].

The practical sphere represents behaviours, management practices, and technical solutions that contribute towards desired outcomes [33,34]. It is often considered the "outcome sphere, where the numbers, parameters, and indicators are most often measured (e.g., the Human Development Index, the Red List of Endangered Species, ecological footprints etc.)" [33] (p.5). The political sphere is important in the context of this research because it cre-

ates the conditions that either enable or disable transformations in the practical sphere [33]. The political sphere includes systems and structures, institutions (laws, regulations, policies), and culture (social norms, codes of conduct, traditions) [32]. In combination, these conditions influence the behaviour of system actors. Given that the political sphere also involves the management of natural systems, such as ecosystems, it is an important sphere on which to focus attention when considering human–carnivore interactions. The personal sphere includes aspects such as individual and collective world views, beliefs, identities, and priorities [33]. World views are defined as a deeply held set of personal beliefs that shape how we perceive the world and guide action [35,36]. Collective worldviews shape the emergent direction to which the system is orientated, thereby constituting the deepest leverage point [28,37]. However, the personal sphere is insufficient on its own to generate fundamental systemic change [29,37].

In this paper, we examine dingo management in Australia by exploring how factors in these spheres constrain or enable the adoption of non-lethal tools and practices through the 'lock-in' of social–ecological systems. System 'lock-in', or 'trap situations', can lead to unsustainable system trajectories. For example, the removal of the dingoes is a major cause of Australia's wave of mammalian extinctions, ecosystem degradation, and livelihood impoverishment [38]. Lock-in traps arise from complex social–ecological interactions in which structures, systems, and the behaviour of individuals reinforce unsustainable choices. Lock-in traps exhibit characteristics of self-reinforcement, persistence, path dependencies, and undesirability [39]. They are thought to emerge from past decisions and events and are reinforced by path dependencies [40]. Abson et al. [32] (p. 35) argue that "much of human action is path dependent, building on the way things have been done previously and relying on established, often institutionalised, knowledge." Smith et al. [8] identified path dependencies as an important factor to explain how wildlife management in Australia became locked into a paradigm of lethal wildlife control, especially in relation to the dingo. Lock-in traps pervade Australian agriculture and impede systemic change towards improving the environmental sustainability of commercial production systems that could lead to necessary system transformation [5,41].

## 2. Materials and Methods

In total, 21 in-depth semi-structured interviews were conducted with stakeholders representing extensive livestock grazing and wildlife conservation. Interviewees were initially identified through purposive sampling to target candidates who met the criteria of having relevant experience or knowledge relating to carnivores or agriculture and willingness to participate in the research [42]. A snowball sampling method enabled the identification and recruitment of additional candidates across a broad spectrum of views relating to human–dingo conflict and coexistence in production landscapes. Interviewees included: livestock producers ($n = 13$) including 6 sheep producers and 7 cattle producers; a livestock industry representative and a representative of the Centre for Invasive Species Solutions (CISS) ($n = 2$); government agencies that oversee wildlife management and/or agricultural interests from New South Wales, Victoria, and South Australia ($n = 3$); staff from environmental and animal protection NGOs ($n = 2$); and a researcher specialising in livestock guardian animals ($n = 1$). Within the sample of Australian livestock producers, eight producers (six cattle and one sheep producer) either used non-lethal tools and/or practices to prevent livestock predation or did not kill dingoes that inhabited or moved through their property (herein non-conventional producers). Five sheep producers and one cattle producer relied primarily on lethal management (shooting, trapping, and use of poisons) to reduce dingo predation (herein conventional producers). Two of the conventional producers participated in a 'pest animal' or 'wild dog' group, that is, a volunteer group of landholders that primarily use lethal control of dingoes, wild dogs, and hybrid offspring.

Interviews were conducted primarily over the telephone and lasted from one to three hours. Interviews commenced with participants introducing themselves and giving an overview of their livestock operation or role in government or an NGO, and their

interactions with dingoes. Livestock producers were questioned about the various ways they mitigate dingo predation on livestock (using lethal or non-lethal tools and methods or a mix of both), the barriers and constraints to adopting tools and practices that are not lethal to dingoes, as well as the processes, events, and conditions which precipitated, facilitated, and enabled adoption to occur (or not). With the participants' consent, all interviews were audio-recorded and transcribed verbatim using a professional transcription service. Interviews were analysed thematically, whereby repeated coding, sorting, and categorising were conducted using the MAXQDA software (VERBI GmbH version 18.2.0) [43]. Exemplary quotes were selected to shed light on recurring themes relating to barriers and how they were overcome in the adoption of preventive innovations.

## 3. Results

### 3.1. Practices to Mitigate Dingo Predation

Lethal means to reduce the risk dingoes pose to livestock (e.g., shooting, trapping, and ground and aerial distribution of poisoned meat baits), were used by five sheep producers, one sheep producer did not use lethal control but instead used a visual predator deterrent called a 'foxlight'. Government representatives reported the use of helicopters to facilitate regional aerial baiting to poison canids such as dingoes, feral dogs, and foxes. Two sheep producers used a mixed strategy that consisted of livestock guardian dogs and donkeys in conjunction with lethal control (i.e., trapping and/or shooting backed up by regional poison baiting).

The seven cattle producers interviewed utilised a mix of lethal and non-lethal approaches or abstained from any action due to the greater resilience of cattle to dingo predation. Six cattle graziers practiced a 'no kill' approach and allowed dingoes to form stable family groups and maintain territories. These graziers reported multiple benefits of reduced livestock injury and loss to predators, less time stalking and killing wildlife, and ecological benefits such as reducing the number of foxes and cats and the dispersal of wild grazers such as goats, rabbits, and kangaroos, which aids pasture mass and livestock profitability. Producers that utilised guardian dogs or identified as organic producers generally refrained from baiting to avoid killing their working dogs or to maintain organic certification. The wide range of approaches to managing real and perceived dingo predation reflects the complexity and heterogeneity of strategies in extensive livestock production systems. The results also revealed interesting insights into the socio-cultural norms influencing human–dingo interactions.

### 3.2. Barriers in the Practical Sphere

Barriers in the practical sphere include lack of landholder capacity (knowledge, skills, and experience) to adopt preventive innovations. Several conventional producers reported difficulties implementing some non-lethal techniques.

*Maremma dogs are very effective, but because there's a lot of work going into them, some people can't handle dogs. You've got to be strict with them and train them or they don't turn out all that good.*

Conventional sheep producer, Victoria

Management difficulties on large landholdings were frequently cited as limiting options for the adoption of non-lethal approaches such as livestock guardian dogs as the livestock can tend to spread out. Landholders may be unable to easily locate and feed guardian dogs, although new GPS tracking technology could overcome those issues. The size and remoteness of many rangeland farming operations in Australia makes the installation, repair, and replacement of equipment very difficult, time-consuming, and expensive. Interviewees reported that these practical barriers to the use of technology in extensive grazing in Australia constrained the implementation of some preventive non-lethal innovations.

Analysis showed that the political and personal spheres exert greater influence on decisions relating to dingo interactions than did the practical sphere. This also aligns with O'Brien's grouping of Meadows' [28] leverage point framework across the three spheres from shallow interventions (practical sphere) to deeper interventions in the political and personal spheres [28,29]. Practical barriers have also been covered in the literature [8]. Accordingly, the results presented here focus on the barriers in the political and personal spheres.

### 3.3. Barriers in the Political Sphere

The following section identifies the most influential barriers in the political sphere including socio-cultural, institutional (laws, polices, information provision and capacity building), and economic factors that constrained the adoption of preventive innovations. The barriers are illustrated by a diversity of stakeholder quotes. Here we explore how and why these barriers impede the adoption of preventive innovations and of systemic change towards coexistence.

#### 3.3.1. Socio-Cultural Barriers

A key barrier to the adoption of preventive innovations is the intense socio-cultural pressure on producers to conform to social norms around lethal dingo control. Pressure to conform to accepted social norms occurs over time from neighbours or peers (i.e., from the bottom up). These norms are reinforced by top-down pressure from industry and government that have, over time, normalised lethal control.

Social pressure to use lethal dingo control was experienced by all the non-conventional producers and took forms such as name-calling, verbal abuse, or feelings of being 'under attack' in their local communities. These socio-cultural factors appeared to be important in creating a self-reinforcing system of lethal dingo control.

> *There is enormous pressure in the neighbourhood. We were mocked and abused. It can get extremely vehement, and it was very, very tough . . . . . . where we were almost considered the downfall of the neighbourhood.*

> Non-conventional cattle producer, South Australia

> *My immediate neighbours are pretty good, but I have had a lot of abuse from other areas.*

> Non-lethal producer, Queensland

Conflicts over dingoes in grazing communities can become extreme and escalate to the point where producers and their families are ostracised from the community simply for refraining from lethal dingo control. It is uncertain why the reaction of conventional producers to non-conventional producers is so extreme.

> *The pressure is definitely there. I mean, we're a bit ostracised from the community but it's hard to ostracise someone that doesn't really care what people think to be perfectly blunt.*

> Non-conventional cattle producer, Western Australia

> *The biggest negative is social interactions with other people because some people are so incensed that I don't do what's always been done [lethal control] and some people have stopped talking to my wife.*

> Non-conventional cattle producer, Queensland

Social pressure can be subversive when people are silenced because they have a view that differs from local social norms that favour lethal dingo control. Producers who spoke out in opposition to lethal control or used non-lethal control practices risked being 'visible'. Non-conventional producers noted that speaking publicly on this issue may make them

a target of social conflict, expressed as an embodiment of the risks (i.e., 'going out on a limb' or 'sticking your head up'). Given the potential to become a target, non-conventional producers needed to be resilient in the face of negative feedback to transcend beyond the norm.

> *It takes a bit of backbone to be able to go out on a limb and make these changes* [adopt non-lethal] *because there are some fairly deeply ingrained views on predators in a lot of regions. So, it takes guts to buck the trend, and try something new.*

> Conservation NGO representative 1

> *There are a few people out there starting to do it* [use non-lethal practices] *but most of them are definitely not prepared to stick their head up at this point because they're going to get a lot of negative impact socially from other people in the area.*

> Non-conventional cattle producer, Queensland

Conventional producers exert pressure on their neighbours to implement lethal control because dingoes are mobile and can travel long distances within large territories. There is a strong narrative that producers must do their part to combat 'pest' species that can negatively impact livestock and livelihoods. However, lethal methods such as baiting can harm livestock guardian and working dogs, which impacts non-conventional producers.

> *People with Maremma guardian dogs, for example don't like using poison, and there are still issues with organic certification. Some people just have an issue with anyone that's accidentally poisoned their kelpie, that it's not visually pleasant, watching a dog or even a fox or canid dying from ingesting a bait ... some people have a moral objection to it.*

> State government representative, Victoria

It appears that conventional producers prefer to use lethal methods because this engenders a sense of control over the 'problem' of predation. There may be a sense of satisfaction to 'get' the perpetrator and a sense that lethal control provides a means to resolve the problem of livestock predation, although it often provides only a temporary 'fix'. An interesting cultural phenomenon is the triumphant display of dead dingoes from a fence or tree like trophies to perhaps gain recognition and appreciation from peers for assisting agricultural communities in dealing with the challenge of dingoes [44].

> *The most popular and favoured control method is leg-hold trapping ... The reason they* [farmers] *like it is, you catch a dog you can hold a dog up by its hind legs and say, here it is, I've got the bastard.*

> State government representative, Victoria

> *You can't hang a dog that's been baited on the fence like you can when it's been trapped ... I call it the 'cricket score mentality', it's all about how many dead dogs we got rather than how few sheep were attacked.*

> Representative of the Centre for Invasive Species Solutions

Culture in grazing industries appears to be heavily influenced by past traditions. Adherence to traditions may create path dependencies that are difficult to deviate from. The social pressure to conform to 'traditional ways', or the lethal status quo, likely arises from the local community. Traditions are reinforced by the livestock industry, creating a negative feedback loop. The traditional values and 'in-built conservatism' in rural communities may contribute to a reluctance to try new tools and approaches, especially when those approaches deviate from practices that have persisted for more than 200 years.

*There is resistance to change, because for generations, things* [lethal control] *that we've been doing for more than a hundred years, is still going on. Because that's how people are taught to do things. And there's sort of this status quo situation.*

Conservation NGO representative 1

*I think it's a case of actually not looking at what's really happening* [on the ground] *just having a theory or a long-held tradition of doing what they do, and not thinking about a better way forward. And I love the thought that tradition is peer pressure from dead people.*

Non-conventional cattle producer, Queensland

3.3.2. Institutional Barriers

As culture and institutions have co-evolved, the prevailing culture of lethal dingo control has influenced the institutional priorities, policies, information, and incentives of both government and the livestock industry. Key actors with political power, such as government agricultural departments and the livestock industry, widely support lethal management of dingoes.

As negative perceptions of dingoes are prevalent in rural communities, this can result in a culture that is largely focused on the eradication of dingoes. Moreover, lethal dingo control is conducted at the local and regional levels often by local government representatives that are drawn from and share the values of conservative rural communities, who often consider dingoes to be a government problem to fix. Thus, the deeply entrenched views that dingoes are a menace to livestock industries and need to be 'controlled' with lethal approaches are reinforced.

*The issue is that the culture is to kill all the dingoes and the government's actions reflect that.*

Non-conventional cattle producer, Western Australia

In most states, landholders are compelled by law to control dingoes by designating them a 'declared pest' or 'threat to biosecurity'. For example, in Victoria:

*In Victoria the Catchment and Land Protection Act requires that all land managers control and where possible eradicate established pest species. Wild dogs and dingoes are a declared pest species when they live in certain areas.*

State government representative, Wild Dog Program, Victoria

This 'declared pest' designation creates an impediment to the use of preventive innovations. As a result, there is a lack of funding and support to investigate these approaches. Landholders who refrain from lethal control face top-down pressure in the form of formal institutional sanctions. They may be subjected to social pressure such as being threatened with legal action for breaking the law.

*I get called names and I am attacked in the media . . . There's also been people trying to get our local council to take legal action against me but that didn't work.*

Non-conventional cattle producer, Queensland

In most Australian states, except Victoria, the management responsibilities for dingoes have devolved to regionally based institutions such as Local Land Services (LLS) in New South Wales, local councils in Queensland, Landscape Boards in South Australia, and Recognised Biosecurity Groups in Western Australia. To create policy and to oversee wild dog groups at a national level, a National Wild Dog Committee was established, comprising industry representatives such as the National Farmers' Federation, Ag Force Queensland,

cattlemen's and graziers' associations, as well as peak industry associations including Australian Wool Innovation (AWI) Limited, Meat and Livestock Australia (MLA), and Wool Producers Australia, as well as state government staff and the Centre for Invasive Species Solutions (https://wilddogplan.org.au/about-the-nwdap/national-committee, accessed on 1 February 2023).

The National Wild Dog Committee developed the National Wild Dog Action Plan 2014–2019, and the revised plan for 2020–2030 that advocates an integrated and coordinated approach to dingoes, referred to as wild dogs [45,46]. These organisations, in turn, provide infrastructure (i.e., systems, structures, and policies) for local community organisations such as 'wild dog groups', 'pest control groups', and 'biosecurity groups' to carry out regionally coordinated dingo control. Membership of wild dog groups is drawn from local farming communities that favour lethal options. Hence, there tends to be an unequal representation or bias in favour of producers who prefer lethal approaches. When lethal control of dingoes becomes the main strategy to reduce predation, it results in a reinforcement of this system, potentially crowding out alternative perspectives, actions, and dissenting voices. The 'nil tenure' approach aims to reduce conflicts between landholders in relation to wildlife. This reinforces a widely held perception that dingoes or wild dogs belong to government.

> *The nil tenure management planning process which was kicked off in New South Wales in 2000 was an approach where taking away the land tenure and the blame for who owned* [wild] *dogs, that was often the case . . . to get rid of those lists of tenures.*

Representative of the Centre for Invasive Species Solutions

> *Bite back* [wild dog] *groups are formed in South Australia to try and coordinate baiting to the Spring and Autumn times and to get people to start working in local collectives to manage the dogs in their regions.*

Conventional cattle producers, South Australia

However, not everyone is satisfied with participating in a wild dog group. One sheep producer who used a mixed strategy of shooting and trapping and livestock guardian dogs to protect his sheep refrained from baiting on his property. Although he did not elaborate as to why he was no longer part of a wild dog group, he made evident that there was some conflict within the group, resulting in his decision to withdraw.

> *We sort of tried to be in that* [wild dog group], *but that's a waste of time, too many people have got too many ideas and there's too many 'blues'* [arguments].

Conventional sheep and cattle producer, Victoria

The livestock industry works in partnership with government to create the infrastructure for poisoned baiting. Government agencies work with established pest groups, providing ready-made baits to landholders. This government support creates a path dependency on poison baiting as the primary strategy for mitigating dingo predation.

> *We have contracts with five Local Land Services in NSW. They can provide a service which is both to the people injecting the baits or putting them out or the aircraft that is distributing. Australian Wool Innovation provides freezers and drying racks so you can dry your meat baits and store them in boxes in the freezers ready for your next baiting program . . . We have technology that's quite adequate, there are no real shortages in technology in terms of disposing of predators, we're actually (in plain language) quite good at killing things.*

Sheep grazing industry representative

*As the coordinator of the* [pest animal] *group, farmers ring me about how many [baits] they want, and I pass that information on. The baits come to the group ready-made* via *Local Land Services.*

Conventional sheep producer, New South Wales

This coordinated approach makes it easy for landholders to deal with dingoes in a lethal way. It also pressures landholders into this behaviour by making it appear almost as a civic duty to communities and the environment. For example, cattle producers had experienced pressure from the government and sheep grazing industry to undertake lethal control despite dingoes not significantly affecting their operations.

*We deal with those cattle producers, because ideally we like them to do wild dog control even though most of them don't need to, because most of the time their cattle's not being eaten.*

State government representative, Wild Dog Program, Victoria

These systems and structures, involving community members in wild dog groups to deliver coordinated control, make lethal control widespread and institutionally supported. This perpetuates the lethal paradigm, creating and reinforcing an institutional lock-in trap.

*I think, one of the limitations here, is that the programs are so widespread, and supported, that there's just a massive resistance to change. So, it's kind of like the machine is too big to influence.*

Conservation NGO representative 1

This maintenance of the status quo is further reified by institutional rhetoric stating that if lethal control were not carried out, jobs would be lost by those who are employed in lethal control activities. In Victoria, the management of dingoes is carried out at the state level under the Wild Dog Program, with a hierarchy of roles including project manager, operations managers, senior wild dog controllers, and trappers colloquially referred to as 'doggers'. The creation of these formal, bureaucratic roles has the potential to create path dependencies to continue investment in and support for lethal control.

*One of the differences in Victoria is that there are about seventeen or eighteen state government employees who are doggers.*

Sheep industry representative

*There's a vested interest by all these people that work in these government programs to have jobs. So, if anybody was to come up with a solution whereas they didn't have to use 1080 poison bait, they wouldn't have jobs.*

Non-conventional sheep producer, New South Wales

Lethal dingo control is 'sold' to landholders as a 'service', and capacity-building activities provided by government largely advocate lethal control techniques. For example, training offered by government agencies to landholders selectively focuses on lethal control, particularily poisoning and trapping. It seems that no commensurate training is provided for the array of preventive innovations. This creates an imbalance in the provision of information about the suite of tools and resources available to manage predation risk and creates a reliance on the 'service' of lethal control.

*We really deliver that frontline delivery service to landholders in doing wild dog controls.*

Government representative, New South Wales

*We'll run days where we have a small group of farmers that are interested in learning how to trap, so we'll demonstrate how you go about trapping. We'll demonstrate the baiting process, why you bait, how you bait, where you bait. We are trying to be more supportive of the guardian animals, particularly Maremmas and Anatolians, just to get a bit of balance, is probably the one thing we fall down on . . . we don't use Maremmas ourselves, we can only trap, bait, shoot and educate.*

State government representative, Wild Dog Program, Victoria

The lack of support for non-lethal options in Australia creates an institutional barrier to change in the political sphere.

*This lack of institutional support is hampering adoption* [of non-lethal methods] *especially when there is much greater support for lethal management using poisoned baits in New South Wales* via *the government agency, Local Land Services, or for trapping* via *the Victorian Government.*

Researcher, Australia

3.3.3. Information Barriers

Industry power reinforces the lock-in to lethal control by setting priorities for information dissemination and research funding that limit preventive innovations. The political power of both the grazing industry bodies, such as Meat and Livestock Australia and Australian Wool Innovation, and chemical industries that manufacture the poison used to kill dingoes stabilises the lethal paradigm. Furthermore, as industry collects producer levies, there is a pool of funding for research and communications which tends to focus on lethal options.

*I think that there's definitely an industry backed research contingent. There's a lot of money that groups like Australian Wool Innovation and others get. So, there is an imbalanced funding. The high power of the wool board they hold the political testicles.*

Conservation NGO representative 2

*Meat and Livestock Australia who provide a lot of the money for the Invasive Animal CRC* [now Centre for Invasive Species Solutions] *decided the only things they were going to fund from now* [in terms of research] *on was anything that killed predators. They were not interested in non-lethal and as far as I know, that hasn't changed since. All those really big corporations basically set the tone for what gets funded.*

Researcher, Australia

The industry messaging appears to purposely devalue dingoes, this is evident in the change of name from dingoes to wild dogs. Lethal control is normalised by devaluing the dingo, referring to it as a 'wild dog' and labelling it an invasive 'pest'.

*The local area we live in, it's had what was called the Dingo Association, now it's the Wild Dog Association and that went right back to the early 1900s.*

Conventional sheep producer, Queensland

Lastly, the media perpetuates a wider social discourse that dingoes are negative and need to be controlled. This media discourse further justifies the continuation of lethal control.

*Whenever there's any article here, television, radio or in print, the bias is inevitably towards getting rid of the dingoes, and of the opinion that maybe it's a bad idea is never voiced . . . It's never a balanced story, ever.*

Non-conventional cattle producer, South Australia

3.3.4. Economic Barriers

The reinforcement of lethal dingo control occurs when livestock producers lobby for institutional support which is provided in the form of subsidised pest control. When industry and government financial support is biased towards lethal options, this creates an economic barrier to the adoption of preventive innovations. There appears to be virtually no institutional support for preventive innovations. Consequently, the costs of adopting preventive innovations are borne by individual landholders.

*Australian Wool Innovation funding has been around 90% on wild dogs and the other 10% on the rest [other native and introduced species].*

Sheep grazing industry representative

*The state government allocates vast majority of its allocated funding is for lethal control* [on private land], *because all the government can do is to control the dogs on Crown Land.*

State government representative, Wild Dog Program, Victoria

Government subsidies for lethal dingo control, such as wild dog bounties in South Australia, are provided under the guise of drought relief. A bounty is a financial incentive or reward, offered by a government for an act or service, in this case the killing of unwanted wildlife [44].

*The bounty program is more a measure of a way of giving back to landholders that are being affected by drought conditions and wild dogs . . . To give them a bit of cash in their pocket that will help them with their cash flow. If they have dogs on their place and they are able to get $120 a dog, then it will help them financially.*

State government representative, South Australia

One producer whose farm was located outside of the dingo barrier fence in South Australia said that such bounties would create an incentive to drive around looking for dingoes/wild dogs to shoot.

*That would have incentivised me to go out and make more of an effort. The amount of times I see dogs and I'm just too busy and I think, no I'm just going to drive past and I don't do anything. But $120 you're definitely going to go looking for dogs. In fact, for $120, it's almost worthwhile driving around having a look for them.*

Conventional cattle producer, South Australia

In summary, it was evident that non-conventional producers experienced various forms of socio-cultural pressure to conform. In addition, the combination of capacity building (e.g., information and training) in lethal methods and financial incentives to kill carnivores (e.g., bounty payments and free or subsidized baiting) has led to a deeply ingrained institutionalisation of lethal control. Despite these significant barriers that are largely of a political nature, there are producers who have transcended these barriers to adopt non-lethal practices.

*3.4. Barriers in the Personal Sphere*

During the interviews, it was clear that conventional producers of small livestock believed that dingoes and sheep were unable to coexist due to the potential for dingoes to injure or kill domestic livestock. Conventional producers appeared to have a reductionist world view that favoured more linear connections between dingo removal and improved farm profit. Intentional and planned killing of dingoes may provide conventional graziers with a sense of control over livestock predation.

*What we can't accept is predation on our production animals.*

Conventional sheep producer, New South Wales

*The only way sheep and dogs can exist is with an exclusion fence, they've got to be separated. If you want to run small animals and be viable* [in business] *and be sane mentally, wild dogs and small animals do not mix. There is no room.*

Conventional sheep producer, Queensland

*We'd like to eradicate them. I don't know if that's possible.*

Conventional sheep producer, New South Wales

*We shoot them when we see them. That's how we control them. Whenever you see one you can shoot, you do.*

Conventional cattle producers, South Australia

Conventional producers had a strong belief in the efficacy and benefit of lethal options. They chose to spend considerable time and resources implementing lethal control despite the recognition that this could create a void for another dingo to occupy. Consequently, lethal control has become a continuously onerous and expensive practice that may lead to short-term benefits but never resolves the root cause of human–dingo conflict.

*We've always got traps set, continuously, because obviously when you remove one dog, it makes room for another one to come in.*

Conventional sheep producer, Queensland

There was a dichotomy between how sheep and cattle producers perceived and interacted with dingoes, with sheep producers having zero tolerance for livestock losses from predators. In general, sheep graziers exhibited a lower tolerance for dingoes than cattle producers. When tolerance for losses is low, this creates impediments to considering options and finding compromises between stakeholder groups that hold different values or knowledge of dingoes. When asked about the efficacy of non-lethal options when compared to lethal options, conventional producers tended to dismiss non-lethal tools and approaches. In contrast, non-conventional graziers were comparatively more open, especially to trialling alternative approaches. Many of them had already gone down this path. Producers who were driven to try a new tool or approach persisted despite the uncertainty and lack of guidance and support.

*Talking to all the producers, that's one of the things they mentioned that it was just extremely hard to get started* [with livestock guardian dogs], *but of course, that's mostly the same for people who actually did persist they had good results.*

Researcher, Australia

Non-conventional graziers also exhibited traits that differentiated them from conventional producers. These included confidence, openness, curiosity, persistence, and less fear of failure if the strategy failed to mitigate livestock predation.

*You've just got to have the confidence, if you try it and it's working you've got to have the confidence to punch through the social barrier.*

Non-conventional cattle producer, Queensland

*The fortunate thing that I've had that's set me aside from all other farmers, even my own family, is the fact that I don't have the fear of failure. You have an idea, if you don't try it then you have nothing. But you have an idea and it fails then you can work your way around it and say, well, if it's got some merit but it's the practical side of it that needs changing.*

Non-conventional sheep and cattle producer, New South Wales

Despite the barriers, several livestock producers transcended the lock-in to transform conflict into coexistence. Non-conventional producers that chose to tolerate or coexist with dingoes appeared to have an expansive worldview that encompassed holistic thinking about the environment and the long-term implications of their decisions. They appeared to prioritise the long term rehabilitation of degraded farmland and ecological sustainability over economic or social considerations. They seemed to possess a strong conviction that their beliefs, values, and actions were appropriate for their farm, which may have enabled them to withstand the intense socio-cultural, institutional, and economic pressure in the political sphere.

*People have said to me, don't you feel like you've got a responsibility to your neighbours and their stock—and I do, but it's nowhere near the responsibility that I have to the Australian people to manage their land well.*

Non-conventional cattle producer, Western Australia

*What matters is the condition of our country and the condition of our stock and what happens in the long term with regards to sustainability and it's going to be very much part of our social license to operate into the future.*

Non-conventional cattle producer, Queensland

## 4. Discussion

Addressing human–carnivore conflict in livestock industries is of global importance [6,11–13,47,48]. This paper elucidates barriers that impede the adoption of non-lethal tools and practices in Australian livestock grazing. In the practical sphere, barriers include a lack of capacity (knowledge, skills, and experience) to implement preventive innovations, the size and remoteness of extensive grazing enterprises, and lack of interest. For example, the use of livestock guardian dogs is seen as difficult across extensive enterprises, although there are examples of it successfully working to reduce predation of sheep on large stations [49]. Lack of interest in non-lethal alternatives arose due to the belief in the efficacy of lethal options, such as shooting, poisoning, and trapping, that are more familiar to producers. Non-lethal alternatives are perceived as yet to be proven effective (e.g., [48]) and potentially costing more. Smith et al. [8] identified various barriers to non-lethal management of dingoes in Australia, including perceived higher cost and lower efficacy of non-lethal compared to lethal control, the size and remoteness of farming operations, lack of government support for non-lethal practices, as well as social stigma. Practical barriers do not represent a reason not to attempt or continue to seek successful approaches. Most of these barriers fall in the practical sphere that on its own has shallow transformation potential [28,29]. However, interventions at shallow leverage points should not be dismissed altogether. There is the potential for ripple effects that create an enabling environment, building and supporting niches of innovation [50,51] that could lead to system transformation [5,52,53].

In contrast to the practical sphere, barriers in the political sphere exerted a powerful influence over system transition towards coexistence with dingoes. These barriers related to socio-cultural, institutional, informational, and economic factors. Iles [41] identified a similar array of political economy and socio-ecological lock-ins that inhibit a transition to agroecology in Australia. As Cocklin and Dibden [54] (p. 4) point out, producers' decisions are not made in isolation, as farmers are "influenced by government policies, which

have until recently been unsympathetic to environmental concerns, and more recently have been largely determined by what might be called the tyranny of the market". To remain profitable, farmers are often forced to clear more native vegetation or overstock their paddocks, despite an awareness that this contributes to land degradation. The influence of the political sphere is not limited to agriculture but also applies to sustainability transformations more broadly. Patterson et al. [55] (p. 2) acknowledge that efforts towards sustainability transformations are likely to be contested politically because "different actors will be affected in different ways and may stand to gain or lose as a result of change." Government and the livestock industry wield significant political power and widely support lethal management of dingoes in the form of policies, information dissemination, capacity building, and financial incentives.

Non-conventional producers appear to share an expansive worldview that incorporates holistic, ecological, and long-term thinking, as well as an openness to new ideas and diverse sources of knowledge about sustainable forms of agriculture. They exhibit a land-stewardship ethic that aligns agroecological principles with a higher tolerance for losses to dingoes and the goals of regenerating landscapes and encouraging species diversity. The willingness to shift worldviews is a fundamental skill enabling entrenched mental models to be transcended [51]. As Ives et al. [37] notes, deep awareness, reflection, empathy, and willingness are required to transcend existing paradigms. For example, some cattle graziers have made a personal decision to cease killing dingoes due to the belief that dingoes are essential for healthy biodiversity and productive agricultural landscapes [23,25,26].

The obvious benefit is reduced persecution of dingoes, yet a stewardship ethic yields multiple environmental benefits [56]. For example, using less poison in the landscape and fewer traps has flow-on benefits for improved animal welfare and landscape health [57]. Kreplins et al. (2018) studied the uptake of poisoned baits for dingo control, the majority of the baits were consumed by non-target species such as varanids, corvids and kangaroos [58]. Furthermore, recent study investigating indices of dingo and fox activity in New South Wales forests found that dingoes had a greater suppressive impact on fox activity than poison baiting, which benefits small mammal biodiversity [59].

Perceptions of identity in the personal sphere appear to influence decision-making. van Eeden et al. [60] found that graziers who identify as environmentalists were less likely to engage in lethal dingo control. Similarly, Naughton-Treves et al. [61] showed that deep-rooted social identity and occupation, such as being a hunter or producer, were powerful predictors of tolerance towards wolves in the United States. Acceptance or rejection of sustainable agriculture as a management philosophy is linked to a personal value system [62]. In effect, non-conventional producers adopt a new paradigm, and this has been recognised as the most powerful tool for transformative change [28,29].

Lock-in traps can impede system transformation and arise from complex social–ecological interactions, in which systems and structures, as well as the behaviour of individuals, reinforce unsustainable choices. This leads to unsustainable outcomes (e.g., loss of biodiversity, degradation of ecosystems, etc.) [38,40]. Others have described lock-in traps as social dilemmas where individual and group benefits are in conflict [63]. Lock-in traps can also arise from human-to-human conflicts about wildlife [64]. The remainder of this discussion elaborates on the four characteristics of lock-in traps identified by Haider et al. [40] (i.e., self-reinforcement, persistence, path dependencies, and undesirability) to shed light on how lethal control of dingoes has become so entrenched in Australia.

*4.1. Self-Reinforcement of a Lethal Paradigm*

Top-down pressure is exerted through formal social norms imposed by institutions (government and the grazing industry). Formal social norms are expressed in laws, policy, preferential knowledge (research funding and available training), and financial incentives (subsidies and bounties) that are all biased towards lethal control. According to Iles [41] (p. 6), once a system or regime becomes stabilised, "it tends to accrete co-evolved, enduring infrastructure, institutions, and behaviours within which actors must operate or live". Our analysis indicates that

factors in the political sphere, such as laws, institutional policy, incentives, and information flows, are aligned with social norms that favour lethal control of dingoes. This finding is echoed by van Eeden et al. [60], who concluded that social norms and policy conditions, such as subsidies and legislation, influence dingo management to such an extent that it focuses almost entirely on encouraging, subsidising, or even mandating lethal control.

Dingoes are framed institutionally as an invasive pest due to their impact on grazing. They are referred to as 'wild dogs' by the pastoral industry and government, despite being considered a native species. The situation is further complicated because dingoes are both a declared pest (legally requiring control) and a protected native species in conservation areas [14,44]. The obligation to control dingoes is reinforced by the National Wild Dog Action Plan, described as a 'livestock-industry driven initiative' that largely dictates the management of dingoes [46]. The agriculture sector wields enormous political power in Australia. For example, the Victorian Farmers' Federation describes itself as "an active, powerful lobby group dedicated to the interests of farmers" (https://www.vff.org.au). Its lobbying and access to government have shaped the construction of environmental and biodiversity laws such as the Wildlife Act 1975 (the Act). Section 7A of the Act states that the governor in council may declare protected wildlife to be unprotected in an area of Victoria [65]. The provision for 'local unprotection' was introduced in 1980, five years after the enactment of the Act [66]. The effect of a species being declared 'unprotected' effectively removes any legal protections for that species under the Act. Unprotected animals such as dingoes can be shot on sight irrespective of whether they have caused damage. Consequently, landholders do not need to apply for an 'Authorisation to Control Wildlife' permit to use lethal control of dingoes in Victoria. This reflects the power of the pest control narrative and the way it has shaped wildlife laws and management in Victoria.

Institutional support for local wild dog groups from the Centre for Invasive Species Solutions and Australian Wool Innovation further amplifies local pressure to conform to lethal control. This institutional support reinforces exclusivity in social capital and ostracises non-conventional producers. The combination of social pressure to conform at the local scale and lack of institutional support (information and incentive schemes promoting alternatives) has ensured that lethal control of dingoes has become ingrained in Australian rural culture for over 200 years. Institutional structures and farming systems have solidified and persist as a lock-in trap.

Non-conventional producers interviewed for this case study faced intense socio-cultural pressure from neighbours and the local community to conform to social norms. Similarly, Johnson and Wallach [7] found there was intense social and legal pressure from neighbouring farms and local governments to conform to conventional lethal practices. Compared to economic or biological factors, social and psychological factors can have a greater influence on behaviour and the uptake of non-lethal interventions [19]. Fear of negative social repercussions and professional isolation [8], as well as the social identity around what it is to be a 'good neighbour' and farmer, also influence producers' decisions to undertake lethal control [60,67]. These factors constitute socio-cultural barriers to the adoption of non-lethal tools and practices.

*4.2. Persistence of the Status Quo*

Systems and structures in the political sphere reinforce the locking in of the status quo (i.e., lethal control of dingoes). Letnic et al. [68] highlight the role of Australian political structures in the maintenance of these practices and the exclusion of non-dominant voices. Van Eeden et al. [67] suggest the status quo has arisen through an over-representation of certain politically powerful interest groups (e.g., hunting and agriculture) in decision-making. Therefore, the interests and lives of dingoes are seen as expendable by powerful groups, and there is a failure to acknowledge the value of dingoes in the Australian landscape and First Nations peoples, as well as a tendency to actively demonise dingoes [69]. Fleming et al. [25] (p. 112) stated that the "adaptive management of wild canids inherently requires compromise and agreement between groups of stakeholders", yet the liberal use

of lethal control implies a marginalisation of stakeholders who see value in coexisting with dingoes. As a result, the voices of researchers, environmentalists, Indigenous Australians, and animal protection groups appear to be excluded from policy decisions [5]. This concentration of the agricultural industry's political power has contributed to the resulting systems and structures (i.e., policies and laws) and further reinforced the paradigm of lethal wildlife management in Australia.

### 4.3. Path Dependencies

Path dependencies can cause rigidities in systems and maintain a lock-in to a particular circumstance or 'solution'. For example, in Australia, wool, beef, dairy, and wheat are key food export commodities [70] inherited because of their introduction during British colonisation. European agricultural practices were transplanted to the Australian continent, despite being less suited to the conditions than Indigenous food and fibres, reflecting historical path dependencies in Australian farming [41]. In the same vein, historical, institutional, and economic path dependencies continue to drive lethal control of dingoes. Governing institutions can be hampered by a lock-in to policies and actions of the past, limiting opportunities to innovate [60,71,72]. This lock-in to past decisions limits the ability of institutions to evolve with and adapt to contemporary societal values such as animal welfare and sustainability. Over time, lethal control has become acceptable, ensuring a lower likelihood of change. Smith et al. [8] argue that the use of lethal control over an extended period has allowed it to become familiar and deeply rooted in culture.

New non-lethal approaches, on the other hand, require producers to acquire new skills, networks, and technology, or make up-front investments that might not be entirely recovered [8]. These barriers can lead to preventive innovations being 'locked out', not because they are not effective but rather due to practical and political factors that make it challenging for producers to adopt them or for policymakers to seriously consider them. Even innovations that are mature (in terms of practice, knowledge, and networks) elsewhere, such as livestock guardian dogs, can founder under these circumstances [41]. This situation stands in stark contrast to farmers' relatively rapid adoption of other improved environmental practices, such as minimum tillage, when encouraged with institutional support [73,74].

Currently, dingo management is controversial and expensive. Economic path dependencies are created when government financial support is available for lethal control but not for alternatives. The results showed how lethal methods (e.g., trapping, poison baiting, bounties) are supported by state government agencies to varying degrees. The South Australian Department of Primary Industries and Regions (PIRSA) promotes baiting, shooting, and trapping as part of an integrated control program inside the dog fence [75]. This includes offering 124,250 commercially manufactured baits to land managers free of charge; a AUD 1.2 million trapper program (between 2018 and 2022) in conjunction with the sheep industry and landscape boards; and a AUD 21 million drought support package that subsidises an additional full-time wild dog trapper, baiting and a AUD 120 bounty for each dead dingo in drought-affected areas [75]. Across all land tenures, including in and around national parks and reserves where dingoes are notionally protected, lethal control and exclusion fencing currently receives the majority of funding.

We found little evidence of training or financial support offered to farmers for preventive, non-lethal methods (e.g., livestock guardian animals or improved animal husbandry). Smith et al. [8] argued that even where preventive innovations have been demonstrated to be effective, as is the case for livestock guardian dogs, uptake has remained relatively low, in part due to lack of government support and incentives.

Our analysis suggests that state subsidies for lethal control, particularly where obscured within other policy objectives (such as drought relief), can result in farmers' expectations of ongoing support as a 'service'. Landholders may become socially conditioned to believe that killing dingoes will automatically improve farm profitability. This creates a cycle of self-reinforcement that is difficult to break. Understanding path dependencies

and how they influence perceptions and knowledge could be a key lever for sustainability transformation [32].

### 4.4. Undesirability

The confluence of social and environmental events contributes profoundly to the creation of lock-in traps [39]. Social–ecological systems, such as commercial livestock production systems, can become caught in both problem-causing and problem-enhancing feedback loops referred to as traps, leading to undesirable social, welfare, ecological, and economic outcomes [63].

Carnivores are sentient and sapient beings that are self-aware and possess rich emotional and cognitive lives [76,77] In addition to direct harm, lethal programs may cause additional suffering through the experience of witnessing individual or social group members being injured and killed [77]. Dingoes have been found to respond to the death of a conspecific in the wild population in similar way to species such as primates, elephants, and some cetaceans [78].

Government-sanctioned lethal methods, such as poisoning and trapping, represent an animal welfare concern because they do not cause an instantaneous death, but can injure, maim, and cause prolonged suffering to both target and non-target species [57]. Meat baits filled with the poison sodium fluoroacetate (1080) are widely distributed to kill canids across Australia with the aim of reducing their impacts on grazing industries [57]. The poison, 1080, is a popular control agent due to its potency, low financial cost, and ease of use (particularly in pre-prepared baits) [79]. Baiting generally occurs twice per year in autumn to kill adult dingoes before they breed, and in spring to kill juvenile dingoes [80]. The humaneness of an animal control method relates to the overall welfare impact that the method has on an individual animal [81]. Sherley [82] developed criteria to determine the humaneness of poisoned baiting that include the speed and mode of action, appearance and behaviour of affected animals, experiences of human victims, long-term effect on survivors, and welfare risk to non-target animals. Based on these criteria, Sherley [82] concluded that 1080 should not be considered a humane poison.

The efficacy of poison baits is also in question in relation to cattle production, with cases showing baiting to be counterproductive to reducing calf loss; losses of calves were reportedly higher in baited areas than in non-baited areas [83–85]. Campbell et al. [84] concluded that "ground baiting, as applied, was ineffective in protecting calves". Disrupting dingo social groups through reducing a pack's size or removing experienced adults that can kill larger, more difficult prey may encourage dingoes to target livestock [86]. Furthermore, a destabilised group may increase reproductive rates and immigration, resulting in a population dominated by juveniles [86]. Livestock loss is not automatically related to dingo abundance [8].

Leg-hold traps are widely used but remain controversial due to negative welfare impacts for target and non-target species. Sharp and Saunders [81] developed an assessment of the humaneness of trapping that includes the degree of physical injury, duration of restraint, method of killing, effects of exposure or dehydration, as well as propensity for anxiety, fear, and stress. A key drawback of traps is they are not target-specific. A variety of native animals, such as wombats, kangaroos, wallabies, brushtail possums, birds, and goannas, have been found in traps intended for canids [87]. Toothed steel-jawed leg-hold traps pose a high risk of serious injuries, including compound fractures, dislocations, and amputations of the trapped limb [87]. Efforts to improve the humaneness of traps led to recommendations for padded steel-jawed traps (e.g., soft catch traps). However, wallabies when caught in soft catch traps continue to suffer serious injuries, including limb dislocation. The clear benefits of preventive non-lethal tools and practices are reductions in the negative welfare and conservation impacts to dingoes and other animals from poisoned baiting and trapping. These potential benefits will remain unrealised until this deadlock can be broken.

## 5. Conclusions

Non-lethal tools and practices constitute an important leverage point for human–wildlife coexistence. However, socio-cultural, institutional, and economic factors, as well as personal worldviews, have caused a lethal paradigm in Australia that stifles innovation and robs farmers of an array of solutions. This lock-in trap creates significant and widescale impediments to the adoption of environmental sustainability in Australian agriculture and hinders conservation of Australia's biodiversity. Using the four characteristics of lock-in traps—self-reinforcement, persistence, path dependencies, and undesirability—our findings shed light on how the lethal control of dingoes has become ingrained in conventional livestock production. Our research provides an in-depth examination of the barriers to coexistence with the dingo in the Australian context and emphasises that greater attention must be paid to the political sphere to overcome this lethal lock-in. Within the political sphere, the lock-in of lethal persecution of dingoes appears to operate predominantly at two scales—informal and local (local socio-cultural norms, peer pressure, etc.), and formal and wide-scale (institutions, policy, capacity-building and financial incentives, etc). Interactions between the two scales (evident in industry lobbying, path dependency, selective information provision, etc.) amplify this cycle of conflict with dingoes in agricultural landscapes. There is a clear need to expand the dingo management toolkit beyond lethal control and exclusion fencing for the benefit of agricultural communities [8,15]. We suggest that to promote the transition to non-lethal approaches, industry bodies and policy makers need to explore opportunities to establish short-term compensation schemes to assist graziers and help cover potential losses, which might otherwise be difficult for producers to accept.

**Author Contributions:** Conceptualization, L.B. and B.J.; formal analysis, L.B.; investigation, L.B.; writing—original draft preparation, L.B.; writing—review and editing, B.J.; supervision, B.J.; funding acquisition, L.B; review and editing B.S. All authors have read and agreed to the published version of the manuscript.

**Funding:** This research was funded by the University of Technology Sydney (UTS) under the UTS Research Excellence Scholarship.

**Institutional Review Board Statement:** The study was conducted in accordance with the Declaration of Helsinki and approved by the Human Resources Ethics Committee of the University of Technology Sydney (ETH18-2568—HREC) in May 2018 for studies involving humans.

**Data Availability Statement:** Not applicable.

**Acknowledgments:** The authors would like to acknowledge the support provided by the Institute for Sustainable Futures, University of Technology Sydney, as well as support and expertise to shape the research provided by Arian Wallach and the reviewers.

**Conflicts of Interest:** The authors declare no conflict of interest. The funders had no role in the design of the study; in the collection, analyses, or interpretation of data; in the writing of the manuscript; or in the decision to publish the results.

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
