# Peer review of "Unlocking Lethal Dingo Management in Australia"

_diversity, doi:10.3390/d15050642_

Round 1

Reviewer 1 Report

As humans spread out of Africa, other species declined in the places they arrived. Two main categories were primarily impacted: big species we could eat and predatory species that competed with us or preyed on us. Wolves and bears greatly declined or disappeared in parts of Europe and North America; large monitor lizards were eradicated in Australia; and so on. The conflict between humans and wildlife (here usually couched in other terms) continues and receives much attention in the literature. Boronyak et al. explore one facet of it, lethal “control” of dingoes in agricultural settings in Australia. The authors interviewed 21 people associated with the process, some in the agricultural (cattle and sheep) sector, others in government and research. They provide a series of quotes that identify social “choke points” such as pressure from others and systemic ones such as funding that “lock in” a lethal response. They make a convincing case that there are multiple challenges standing in the way of adoption of alternative management options. A long Discussion follows to suggest some ways around such blockages. Personally, I tend to agree with the views of the authors, telegraphed quite early when they approvingly call some land-owners “innovative producers.” Alas, I think they make their case rather badly.

Most of my concerns have to do with the writing. The authors “know what is right” (section 4.4), but they do a poor job of convincing anyone who isn’t already in their camp. We are told repeatedly that many of the people they interviewed do not accept alternative approaches, but there is no concerted effort to tell us what those are and convince us that they work, let alone offer a superior alternative to lethal control. At the same time, the problems with the killing are not really discussed until the very end. In what way are the “innovative” approaches better? There is mention of some of the concerns with approaches like dogs, which are perceived as complex to deploy, but no real answer to such concerns. So yes, ranchers find it difficult to switch – but why would they want to? The authors do not answer that question in a way that anyone who doesn’t already agree with them would find convincing.

Another drawback, also in the writing, is lack of reference to the extensive literature from outside of the Australian dingo realm. As just one example, a study from Costa Rica (Amit et al. Jaguar and puma attacks on livestock in Costa Rica. Human–Wildlife Interactions 7(1):77–84, Spring 2013). For another, a recent book (Angelici and Rossi, 2020, Problematic Wildlife II) is not even cited once. Are there relevant lessons to learn from situations anywhere else? Or perhaps ones that would benefit from the perspective presented here? Not only is this broader perspective not presented, but it feels like the authors may not be familiar with it. Their passion definitely comes through, but is not enough.

Perhaps related to that, Diversity is billed as a “journal on the science of biodiversity from molecules, genes, populations, and species, to ecosystems.” To me, this is a study on human-wildlife conflict, focused on agricultural production and advocating alternative ethical approaches, that has very little to do with diversity. Again, I tend to agree with the authors that lethal control generally offers few benefits, and has negative impacts to non-target species – but they left me completely unconvinced about that here.

Finally, a methodological concern. Although the interviews held were quite lengthy (1-3 hours each), there were only 21 of them. Worse, since the interviewees helped identify other interviewees (l 122), it is unclear to me how many independent perspectives are presented, and therefore how representative the findings are. What they report comports with what I hear from people half the world away, so I’m inclined to believe the findings are meaningful – but my gut feelings are not a good guide and actual data are required. This, too, is a place where a broader perspective could help solidify the conclusions.

Reviewer 2 Report

The paper is well written, and covers a very interesting topic; the barriers to the uptake of non-lethal control methods with regards to dingoes in Australia. This is a topic that warrants a lot more attention in my opinion, and it is nice to see a paper looking at this topic in a bit of detail. I really enjoyed reading it, and the authors have done a good job in their data analysis and presentation of results. As I mentioned, the paper is really well written, although in my opinion quite lengthy, and particularly the discussion could benefit from being shortened a bit to make the text more concise. Other than that, I only have a few comments on specific sections:

Page 2, lines 47-49. The adoption . priorities. This needs a reference.

The introduction is good, and has all the information to set up the study. But I am missing a paragraph at the end stating what this particular paper will be looking at, and will be investigating.

Throughout the results sections quotes are used to showcase the point the authors are making, except for the practical sphere sections. I understand that this sphere is considered less influential than the others, but it might be good for consistency in presentation of results to have some quotes there as well.

Reviewer 3 Report

The study has merit in providing an understanding of and pathways to the resolution of human-wildlife conflict. It addresses the most vexatious exemplar of this issue, namely predation by wild carnivores on livestock. The focus is the canid, the dingo, and its management in relation to the production of sheep (goats) and cattle in Australian rangelands. This is complex as the dingo is a relatively recent migrant to Australia (~5,000 y) and likely displaced an endemic carnivore, the thylacine. Furthermore, the dingo will hybridise with more recently introduced domestic dogs leading to some confusion as to the status of such progeny. To add to the complexity of dogs in the rural areas, a culture of hunting feral pigs with large breeds of dogs has lead to unintended release into the pool of feral canids that can threaten rural communities and their enterprises as occurs in regions such as the greater Darwin rural area.

Thus I have some issues with this paper in regard to definitional clarity. The authors address lethal dingo management but need to be clearer about their definition of a dingo and the separation of its management from a broader problem with feral canids. The sample used to understand the wildlife conflict is small and likely not definitive of the Australian community. With this, some latitude must be given. The majority urban community is likely to take a principled stance against lethal dingo management while acceding to local council by-laws that may significantly constrain the number and behaviour of their canid companions. The minority rural community is likely to take a practical stance and in many instances support lethal dingo management for reasons well-summarised in this paper.

A further definitional issue I have with the paper is the use, if not over-use, of the term ‘innovation’.  The authors canvas non-lethal control against lethal control but suggest the former is innovative. This gives the impression of a likely unintended bias in the authors approach to the issue. The non-lethal methods described are well-founded and so not innovative. The users did not invent them but have been progressive in adapting them to their individual circumstances. The term ‘preventative innovations’ is a mouthful and unhelpful. The authors are simply contrasting the application and uptake of one of two alternatives, lethal and non-lethal control. The description of the users of non-lethal control as innovators is overreach since their behaviour is clearly multidimensional in their ‘eco-friendly’ management but the paper is about one dimension, lethality of dingo control. Furthermore, the authors admit in the discussion that ‘conservatives’ can be innovators (progressive) in adopting farming practices while exercising lethal control of dingoes. Thus I would suggest dropping innovation from the discourse and simply reference two alternatives, lethal and non-lethal control, and in application conservative and non-conservative practitioners. Conservatism needs clear definition in the methods.

Finally, I would be wary of emotive terms like dingo persecution (line 815) but I have no problem that dingoes may be ‘demonised’ in discussion of their impacts.

The paper is very well-written. The discussion is quite lengthy, and the authors may need to review this for unnecessary repetition.

Reviewer 4 Report

see attached
